# Elevation-Dependent Contribution of the Response and Sensitivity of Vegetation Greenness to Hydrothermal Conditions on the Grasslands of Tibet Plateau from 2000 to 2021

Yatang Wu [1], Changliang Shao [2], Jing Zhang [3], Yiliang Liu [3], Han Li [3], Leichao Ma [4], Ming Li [4], Beibei Shen [5], Lulu Hou [2], Shiyang Chen [2], Dawei Xu [2], Xiaoping Xin [2,*] and Xiaoni Liu [1]

1. Key Laboratory of Grassland Ecosystem, Ministry of Education, Sino-U.S. Centers for Grazing Land Ecosystem Sustainability, Ministry of Science and Technology, Pratacultural Engineering Laboratory of Gansu Province, Pratacultural College, Gansu Agricultural University, Lanzhou 730070, China; wuyatang666@163.com (Y.W.); liuxn@gsau.edu.cn (X.L.)
2. State Key Laboratory of Efficient Utilization of Arid and Semi-Arid Arable Land in Northern China, National Hulunber Grassland Ecosystem Observation and Research Station, Institute of Agricultural Resources and Regional Planning, Chinese Academy of Agricultural Sciences, Beijing 100081, China; shaochangliang@caas.cn (C.S.); houll94@163.com (L.H.); genghisyang@outlook.com (S.C.); xudawei@caas.cn (D.X.)
3. National Remote Sensing Center of China, No. 8A Liulinguan Nanli, Haidian District, Beijing 100036, China; zhangjing@nrscc.gov.cn (J.Z.); liuyiliang@nrscc.gov.cn (Y.L.); lihan@nrscc.gov.cn (H.L.)
4. Natural Resources Comprehensive Survey Command Center, China Geological Survey, Beijing 100055, China; laozhuangboy@sohu.com (L.M.); lm18910077797@163.com (M.L.)
5. Aerospace Science and Industry (Beijing) Spatial Information Application Co., Ltd., Beijing 100070, China; 82101191163@caas.cn
* Correspondence: xinxiaoping@caas.cn

**Abstract:** The interrelation between grassland vegetation greenness and hydrothermal conditions on the Tibetan Plateau demonstrates a significant correlation. However, understanding the spatial patterns and the degree of this correlation, especially in relation to minimum and maximum air temperatures across various vertical gradient zones of the Plateau, necessitates further examination. Utilizing the normalized difference phenology index (NDPI) and considering four distinct hydrothermal conditions (minimum, maximum, mean temperature, and precipitation) during the growing season, an analysis was conducted on the correlation of NDPI with hydrothermal conditions across plateau elevations from 2000 to 2021. Results indicate that the correlation between vegetation greenness and hydrothermal conditions on the Tibetan Plateau grasslands is spatially varied. There is a pronounced negative correlation of greenness to maximum temperature and precipitation in the northeastern plateau, while areas exhibit stronger positive correlations to mean temperature. Additionally, as elevation increases, the positive correlation and sensitivity of alpine grassland vegetation greenness to minimum temperature significantly intensify, contrary to the effects observed with maximum temperature. The correlations between greenness and mean temperature in relation to elevational changes primarily exhibit a unimodal pattern across the Tibetan Plateau. These findings emphasize that the correlation and sensitivity of grassland vegetation greenness to hydrothermal conditions are both elevation-dependent and spatially distinct.

**Keywords:** alpine grassland; elevation; greenness; climate change

## 1. Introduction

Alpine grasslands represent the predominant and distinct ecosystem of the Tibetan Plateau, crucial in $CO_2$ sequestration and sustaining ecosystem equilibrium [1–4], with implications in global climate change mitigation [5,6]. Characterized by unique hydrothermal conditions (notably low temperatures and restricted precipitation), these grasslands inhabit

sensitive eco-climatic zones with pronounced elevation gradients and minimal anthropogenic disturbances, making them one of the most pronounced indicators of warming and early warning systems on the Earth's surface [7–11]. Concurrently, vegetation greenness serves as a robust marker for climate sensitivity within this ecosystem [12]. The dynamic alterations in greenness exhibit a pronounced correlation with hydrothermal conditions, with geographical terrain, particularly elevation gradients, exerting significant influence [11,13–16]. Thus, Alpine grasslands offer an optimal setting for investigating the interplay between vegetation greenness dynamics and hydrothermal conditions, modulated by elevation.

Remote sensing (RS) offers the benefits of efficiency, extensive coverage, and diverse information, establishing itself as the most effective tool for monitoring vegetation greenness dynamics at large range [6]. This technique provides a wealth of data for analyzing climatic responses and sensitivities [17]. The advancements in RS have enabled instruments such as the Moderate Resolution Imaging Spectroradiometer (MODIS) to offer more expansive coverage, superior temporal resolution, and cost-effective, freely available data. Moreover, MODIS provides a range of vegetation reflectances [6,12,18,19]. Over the past few decades, vegetation indices (VIs) derived from these reflectances have emerged as reliable markers of photosynthetic activity. They adeptly capture the responsiveness and sensitivity of vegetation greenness to climatic shifts [3,14–16,20]. Most VIs, including the widely recognized Normalized Difference Vegetation Index (NDVI), are formulated by integrating the red and near-infrared bands [21]. As a predominant indicator, NDVI exhibits high sensitivity to grassland growth conditions and has been effectively utilized to explore the implications of climate change on grassland ecosystems across diverse spatiotemporal frameworks [16,17,20,22].

The accuracy of NDVI is subject to scrutiny due to its tendency to saturate in areas of dense vegetation and its susceptibility to soil backgrounds, canopy brightness, and shadows when coverage falls below 50% [23,24]. Additionally, atmospheric disturbances, such as aerosols, frequently introduce noise into images generated from red and near-infrared bands [25]. It is important to note that grasslands typically exhibit lower vegetation coverage and smaller canopy dimensions, coupled with increased spatial heterogeneity. This often leads to satellite imagery comprising more bare soil pixels relative to other ecosystems [26]. Nevertheless, Wang et al. [18] have introduced the NDPI, an innovative vegetation index designed to discern the difference between green vegetation and soil backgrounds while attenuating this differentiation. Employing a weighted shortwave infrared (SWIR) band instead of the red band in NDVI, NDPI is sensitive to vegetation water content, enabling it to track variations in canopy water content [18]. Crucially, NDPI merges these two functionalities into a singular VI without compromising its sensitivity to vegetation greenness [18]. Xu et al. [27] demonstrated NDPI's superior capability in estimating aboveground fresh biomass across expansive grassland regions with pronounced spatial heterogeneity, particularly in reducing the interference of soil background in Inner Mongolia grasslands. In conclusion, NDPI presents itself as a promising tool for assessing the greenness of Alpine grassland vegetation.

Numerous studies have investigated the response of vegetation greenness to climate change at both regional and global scales. For instance, Xu et al. [28] demonstrated that NDVI changes were significantly correlated with annual temperature (R = 0.52, $p < 0.01$) across China but not with annual precipitation ($p > 0.1$). Additionally, correlations between vegetation greenness changes and both temperature and precipitation were found to be significant at a regional scale ($p < 0.001$). However, many of these studies primarily focus on the annual mean temperature, neglecting the potential influences of minimum and maximum temperatures during the growing season. Previous research has also highlighted the significant impact of elevation on the variations in Alpine vegetation greenness in response to hydrothermal conditions on the Tibetan Plateau [11,14–16]. Specifically, An et al. [14] identified a pronounced elevation-dependent relationship between vegetation greenness and temperature during the growing season (May–September) from 2000 to 2016. Furthermore, Wang et al. [16] noted a mismatch between the elevational variation rate of NDVI and hydrothermal conditions. At

altitudes above 2400 m, temperature predominantly influenced the elevational shifts of NDVI isolines, whereas precipitation was the dominant factor below 2400 m. Wang et al. [16] also observed that the drought response (SPI/SPEI) of the Enhanced Vegetation Index (EVI) significantly diminished with increasing elevation ($p < 0.001$). Despite these findings, the specific influence of elevation on the response and sensitivity of grassland vegetation greenness to minimum/maximum temperatures during the Tibetan Plateau's growing season remains unexplored. This study seeks to address this gap.

This study addresses a current gap in the literature by (1) examining the sensitivity of vegetation greenness to hydrothermal conditions, encompassing minimum, maximum, and mean temperatures, as well as precipitation, on the Tibetan Plateau grasslands from 2000 to 2021, and (2) assessing the impact of hydrothermal factors on grassland vegetation greenness across various vertical gradient zones of the Plateau. A comprehensive understanding of how grassland vegetation greenness dynamics respond to hydrothermal conditions, modulated by elevation, is imperative. Such knowledge will offer valuable scientific insights for sustainable grassland management and precise restoration in ecologically vulnerable zones, particularly in regions characterized by significant elevational variations.

## 2. Materials and Methods

### 2.1. Study Region

The study region, spanning $26°00'12''$–$39°46'50''$N and $73°18'52''$–$104°46'59''$E (as shown in Figure 1a), is situated in southwestern China, predominantly characterized by Alpine grasslands, the most prevalent vegetation type in the area. Enveloped by towering mountains, the region displays an extensive vertical distribution, which results in a distinct nonzonal Alpine climate. This climate is typified by low temperatures, with the majority of the area registering a mean annual temperature of less than 0 °C, further intensifying with elevation and exhibiting pronounced susceptibility to warming [10]. Notably, a significant portion of the region experiences an arid or semiarid climate, with mean annual precipitation falling below 400 mm for over half the area. Rainfall predominantly occurs during the growing season, spanning May to September. The average elevation exceeds 4000 m (illustrated in Figure 1b), rendering the plateau both colder and more variable than other regions sharing the same latitude. Amidst the overarching theme of global climate change, recent decades in this area have seen discernable climatic shifts: a marked acceleration in warming, coupled with increased and variable precipitation patterns [29]. Given these unique characteristics, this region presents an optimal setting for examining the dynamics of grassland vegetation greenness and its responsiveness to climate fluctuations. As per the Chinese Grassland Classification System [30], the primary grassland categories found here include Alpine meadow, Alpine steppe, and Alpine desert (refer to Figure 1c).

### 2.2. Dataset Processing

The NDPI is described by Equation (1). In this equation, RED, NIR, and SWIR represent the surface reflectance values in the red, near-infrared, and shortwave infrared bands (~1.6 μm), respectively. These reflectance data are derived from the MODIS surface reflectance product (MOD09A1 V006) with a 500 m spatial resolution, offering 8-day composite values of normalized reflectance. This product is sourced from the U.S. National Aeronautics and Space Administration (NASA) (accessible at http://earthdata.nasa.gov/, accessed on 2 March 2023). The MOD09A1 was selected to determine the vegetation dynamics across the study region, primarily due to its capability to filter out undesirable values, cloud contaminants, and other data artifacts during processing, ensuring data quality. For data management, the MODIS reprojection tool (MRT) software (version 41) facilitated the mosaicking of eight MODIS tiles every 8 days. This tool also projected the surface reflectance data for the entirety of the study area during the growing seasons (May to September) from 2000 to 2021, employing the Albers map projection (stored in Geo-Tiff format). Ultimately, the maximum-value compositing method was utilized to compile the 8-day NDPI data, resulting in a synthesized value for the entire growing season.

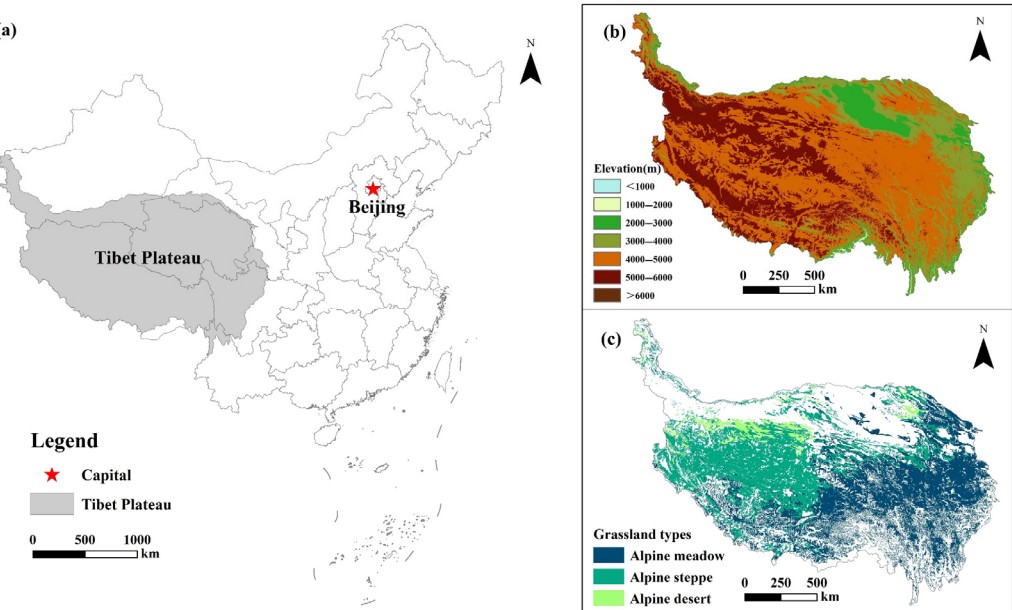

**Figure 1.** Geographical location (**a**), elevation (**b**),grassland type (**c**) of the study area.

$$\text{NDPI} = \frac{\text{NIR} - (0.74 \times \text{RED} + 0.26 \times \text{SWIR})}{\text{NIR} + (0.74 \times \text{RED} + 0.26 \times \text{SWIR})} \tag{1}$$

The minimum temperature ($T_{min}$, °C), maximum temperature ($T_{max}$, °C), mean temperature ($T_{mean}$, °C), and mean precipitation (GSAP, mm) during the growing season were acquired and processed using the A Big Earth Data Platform for Three Poles (http://poles.tpdc.ac.cn/zh-hans/, accessed on 3 March 2023) [31,32] and the National Tibetan Plateau/Third Pole Environment Data Center (https://data.tpdc.ac.cn/home, accessed on 3 March 2023) [33–37]. Data extraction was confined to our study area by utilizing the vector boundary of the Tibetan Plateau. This boundary dataset was sourced from the National Geomatics Center of China (http://ngcc.sbsm.gov.cn/, accessed on 3 March 2023).

The Digital Elevation Model (DEM) with a global resolution of 90 m was sourced from the Shuttle Radar Topography Mission (SRTM) images (available at http://srtm.csi.cgiar.org, accessed on 3 March 2023). These data were utilized to represent elevation characteristics across the Tibetan Plateau. To ensure consistency with other datasets, the DEM was resampled to a 500 m resolution. Additionally, the Albers Equal Area projection was selected for this analysis. Subsequently, these resampled grids were employed for the study.

### 2.3. Methods

The spatial distribution map representing the mean grassland NDPI during the peak of the growing season from 2000 to 2021 was generated by calculating the average value across the entire study area for each of the 22 years. The coefficient of variation (CV) of NDPI can serve as an indicator of stability [38]. The trend and significance of NDPI were discerned, using the Theil–Sen (TS) method combined with the non-parametric rank-based Mann–Kendall (MK) test [39,40].

To evaluate the interannual variations in the maximum greenness of grassland vegetation in response to $T_{min}$ during the growing season, partial correlation coefficients were computed between the NDPI and $T_{min}$, considering $T_{max}$, $T_{mean}$, and GSAP as control variables [12]. The apparent sensitivity of NDPI to $T_{min}$ was quantified by the coefficient derived from multiple linear regressions, where the NDPI was regressed against $T_{min}$, $T_{max}$, $T_{mean}$, and GSAP [12]. Similarly, the response and sensitivity of the NDPI to $T_{max}$, $T_{mean}$, and GSAP were assessed.

## 3. Results

### 3.1. Spatial Patterns of Peak Season NDPI and Hydrothermal Factor Trends

On a grid cell scale, marked heterogeneity is evident in the spatial patterns of the annual mean grassland NDPI during the peak season spanning 22 years from 2000 to 2021 within the study area (Figure 2a). The results indicate that the general spatial distribution of the annual mean NDPI exhibited an incremental trend from the northwestern to the southeastern regions of the study area. Notably, areas with the lowest values (mean NDPI < 0.3), indicating sub-optimal vegetation photosynthetic activity, were predominantly located in the northwestern region of the plateau. Conversely, regions with higher NDPI values (mean NDPI > 0.6), signifying robust vegetation photosynthetic activity, were primarily concentrated in the eastern and northeastern regions of the plateau. The coefficient of variation (CV) highlights significant NDPI fluctuations in the southern region of the plateau from 2000 to 2021 (Figure 2b).

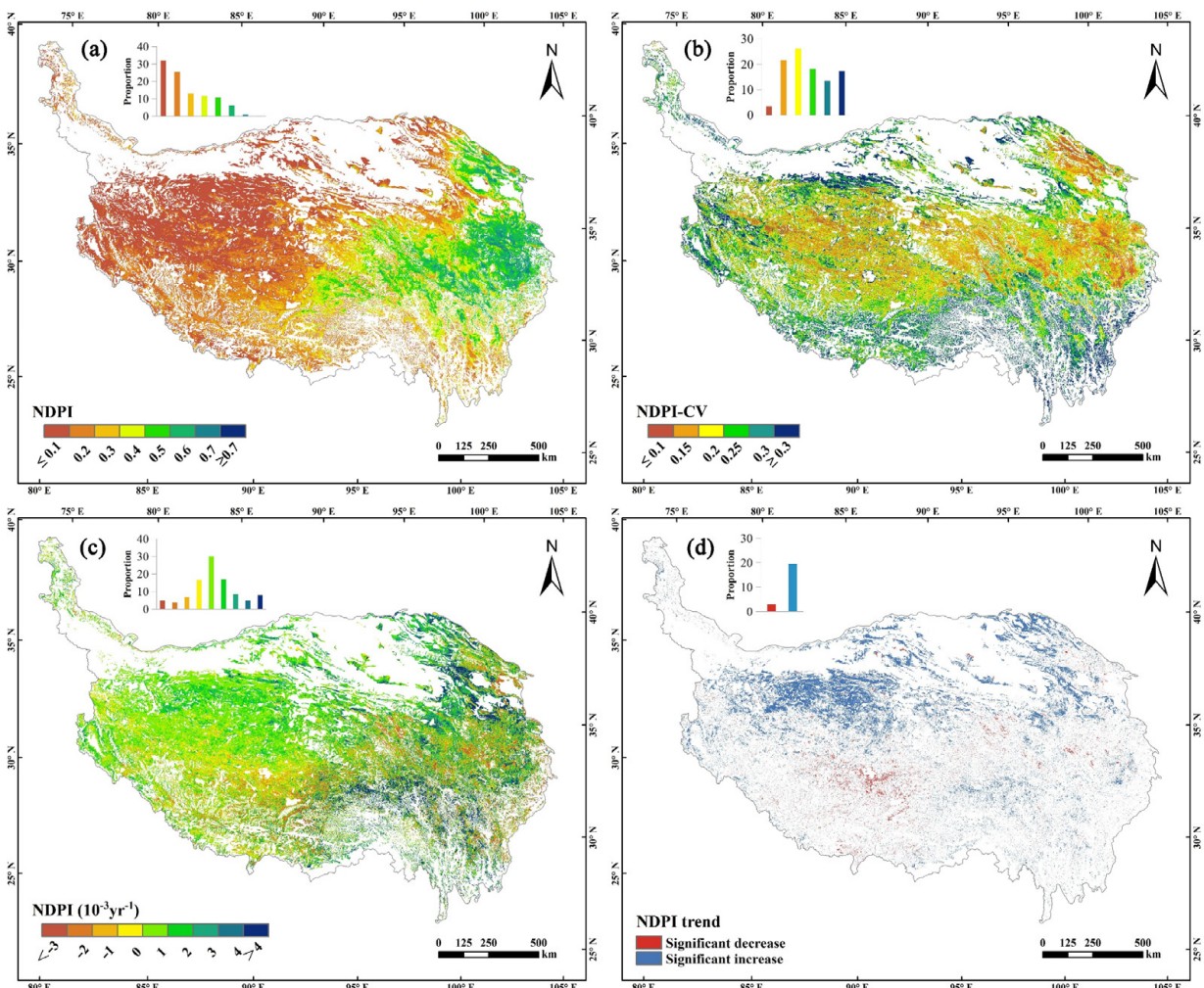

**Figure 2.** Mean value (**a**), stability (**b**), trends (**c**) and significance of the trends (**d**) of NDPI in the peak season over the period 2000–2021.

Figure 2c reveals the spatial distribution of the overall trends in annual grassland NDPI within the study area during the peak season over the past 22 years. A statistical analysis of the grid cells indicates that 67.92% displayed an upward trend in annual grassland NDPI. Of these, 19.57% exhibited a significant increasing trend with $p < 0.05$ (Figure 2d and Table 1). These cells predominantly occur in the northern and northwestern regions of the Plateau, specifically in counties like Gêrzê and Nyima. In contrast, the remaining cells demonstrated a downward trend (slope < 0), with a mere 2.97% showing a significant decrease ($p < 0.05$). These are primarily dispersed across the southwestern

regions of the plateau, including Shenzha and Baingoin counties. Over this 22-year period, trends varied across different grassland types. Alpine meadows, for instance, witnessed a higher percentage of vegetation greenness increases compared to Alpine steppes and deserts. Specifically, Alpine meadows comprised 52.79% and 34.53% of the grid cells with significant increases ($p < 0.05$) and minor increases ($p > 0.05$), respectively. This is followed by Alpine steppe (25.54% and 47.01%) and desert (11.79% and 50.68%) (Table 1).

**Table 1.** The area percentage of the significance of NDPI trend for different grassland types.

| Grassland Types | Area Proportion of the Significance of NDPI(%) | | | |
|---|---|---|---|---|
| | SD | NSD | NSI | SI |
| Alpine meadow | 1.34 | 11.34 | 34.53 | 52.79 |
| Alpine steppe | 3.06 | 24.38 | 47.01 | 25.54 |
| Alpine desert | 3.05 | 34.48 | 50.68 | 11.79 |
| All | 2.97 | 29.11 | 48.35 | 19.57 |

Note: SD, NSD represent the NDPI decrease at $p < 0.05$, $p > 0.05$, respectively. SI, ESI represent the increase at $p > 0.05$, $p < 0.05$, respectively.

Figure 3 presents the spatial distribution and trends of $T_{min}$, $T_{max}$, $T_{mean}$, and GSAP. Overall, $T_{min}$ demonstrated an incremental trend from the northwestern to southeastern regions between 2000 and 2021. The northwestern plateau, particularly in the Ngari and Nagqu Prefecture, registered the lowest $T_{min}$ values. Conversely, the highest $T_{min}$ was observed in the northern region (Figure 3a). Both $T_{max}$ and $T_{mean}$ depicted comparable spatial patterns, indicating a progressive rise from the center towards the periphery. The Golmud region recorded the lowest values for $T_{max}$ and $T_{mean}$ (Figure 3b,c). GSAP displayed a decreasing pattern from southeast to northwest. Specifically, the Alpine desert in the northwestern plateau had GSAP values below 80 mm, whereas the southeastern edge exhibited values exceeding 560 mm (Figure 3d). A notable 64.38% of grid cells presented an annual declining trend, with a decrease rate in $T_{min}$ exceeding 0.03 °C/year, particularly in the northern and northwestern regions (Figure 3e). Contrarily, a pronounced increasing trend of $T_{min}$, surpassing 0.03 °C/year, was evident in the plateau's eastern region (Figure 3e). A significant 80.42% of grid cells demonstrated a decreasing trend annually, with $T_{max}$ decline rates surpassing 0.03 °C/year in the northwestern plateau (Figure 3f). Intriguingly, 78.64% of grid cells displayed an annual increase in $T_{mean}$, with a rate exceeding 0.06 °C/year in the southwestern region (Figure 3g). However, pronounced decreasing trends in $T_{mean}$, less than $-0.03$ °C/year, were observed in the northwestern plateau (Figure 3g). Over half the grid cells (54.43%) exhibited a decadal increasing trend, with the GSAP growth rate exceeding 4 mm/year in the eastern plateau (Figure 3h). Nonetheless, significant declining trends in GSAP, less than $-2$ mm/year, were identified on the southeastern frontier of the study area (Figure 3h).

### 3.2. Hydrothermal Response and Sensitivity of NDPI

The analysis of partial correlations with hydrothermal conditions, specifically $T_{min}$, $T_{max}$, $T_{mean}$, and GSAP, spanned a period of 22 years on a pixel scale. The delineated spatial patterns emphasize functional discrepancies (Figure 4a,c,e,g). Over the past 22 years, a predominant negative correlation between NDPI and $T_{min}$ covered the majority of the plateau, accounting for 58.49% of all grid cells. However, only around 1.89% of these cells exhibited a marginally significant negative correlation at $p < 0.05$ (Figures 4a and 5). Conversely, a positive correlation between NDPI and $T_{min}$ was statistically significant in nearly 1.35% of the grid cells from 2000 to 2021 (Figures 4a and 5). Akin to $T_{min}$, the NDPI negative response to $T_{max}$ was observed in 55.06% of grid cells over the 2000–2021 timeframe. Notably, these negative correlations primarily manifested in the northeastern and western sectors of the plateau, with 8.93% being significant at $p < 0.05$ (Figures 4b and 5). For $T_{mean}$, a positive correlation was evident in 62.13% of the grid cells, primarily in the central and eastern regions. Of these, 11.27% were statistically significant at $p < 0.05$ (Figures 4c and 5). In a pattern analogous to $T_{mean}$, the NDPI's positive

response to GSAP was evident in 58.25% of the grid cells, with the central plateau being the main region of observation. Here, 6.04% of the cells showcased significance at the $p < 0.05$ threshold (Figures 4d and 5).

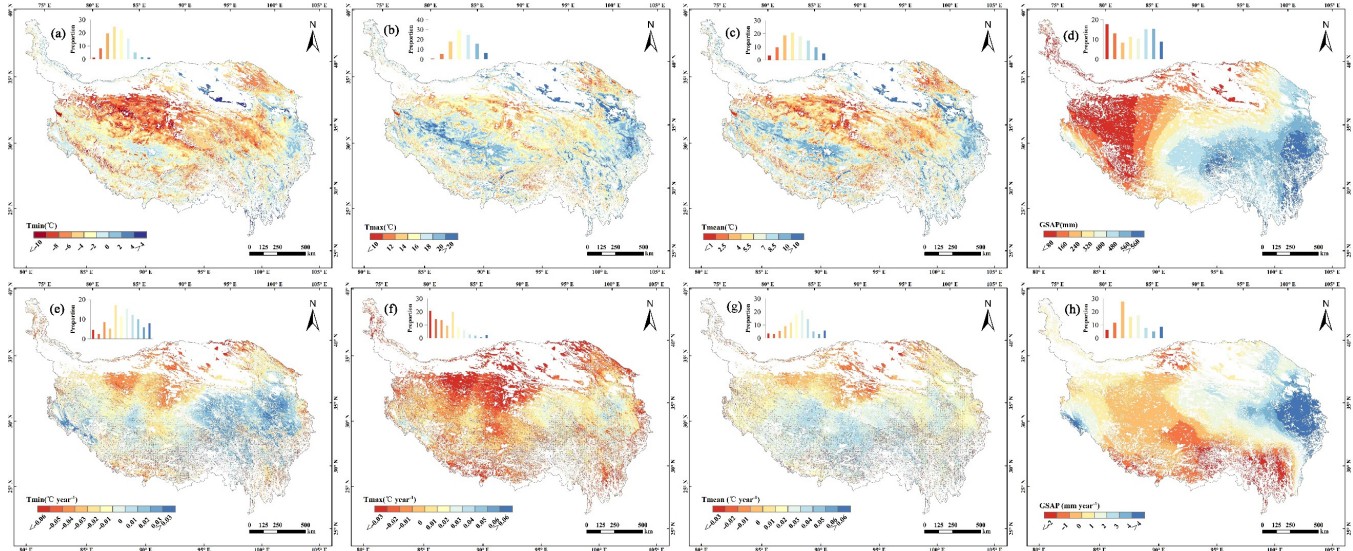

**Figure 3.** Magnitude and trend in hydrothermal conditions on the Tibet plateau. The magnitude of $T_{min}$ (°C) (**a**) $T_{max}$ (°C) (**b**), $T_{mean}$ (°C) (**c**) and GSAP (mm) (**d**) over the period 2000–2021; the trend of $T_{min}$ (°C year$^{-1}$) (**e**), $T_{max}$ (°C year$^{-1}$) (**f**) $T_{mean}$ (°C year$^{-1}$) (**g**) and GSAP (mm year$^{-1}$) (**h**) over the period 2000–2021.

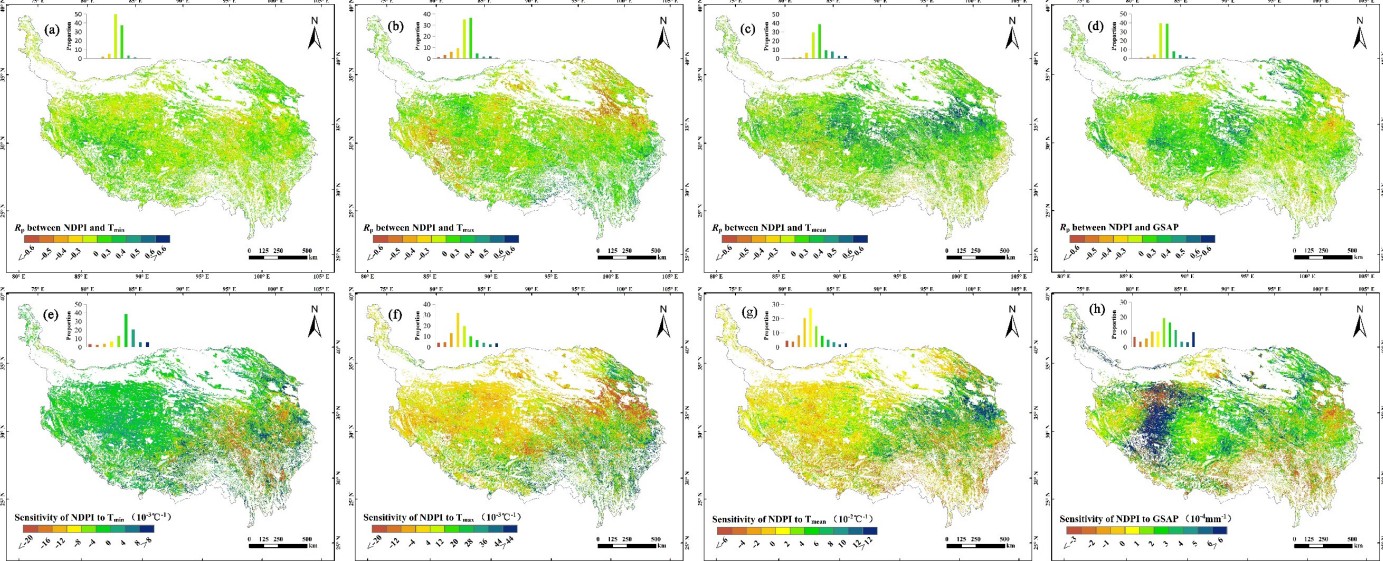

**Figure 4.** The spatial patterns of the hydrothermal response and sensitivity of NDPI. The partial correlation between NDPI and $T_{min}$ (**a**), $T_{max}$ (**b**), $T_{mean}$ (**c**) and GSAP (**d**), respectively; the sensitivity of NDPI to $T_{min}$ (**e**) and $T_{max}$ (**f**), $T_{mean}$ (**g**) and GSAP (**h**), respectively.

The partial correlations between NDPI and hydrothermal conditions were further assessed across various grassland types. Figure 5 depicts the percentage area of partial correlations between NDPI and $T_{min}$, $T_{max}$, $T_{mean}$, and GSAP for each of the three distinct grassland types. The data indicated that the Alpine desert had the highest percentage area showing a negative response of NDPI to $T_{min}$ ($R_P < 0$), succeeded by the Alpine steppe and Alpine meadow (Figure 5). For $T_{max}$, the Alpine steppe exhibited the most significant negative response of NDPI ($R_P < 0$), with the Alpine desert and Alpine meadow following

suit (Figure 5). Conversely, the Alpine steppe displayed the highest percentage area of a positive NDPI response to $T_{mean}$ ($R_p > 0$), trailed by the Alpine meadow and Alpine desert (Figure 5). Comparatively, the Alpine steppe dominated in terms of the positive NDPI response to GSAP ($p < 0.05$), with subsequent rankings being the Alpine desert and Alpine meadow (Figure 5).

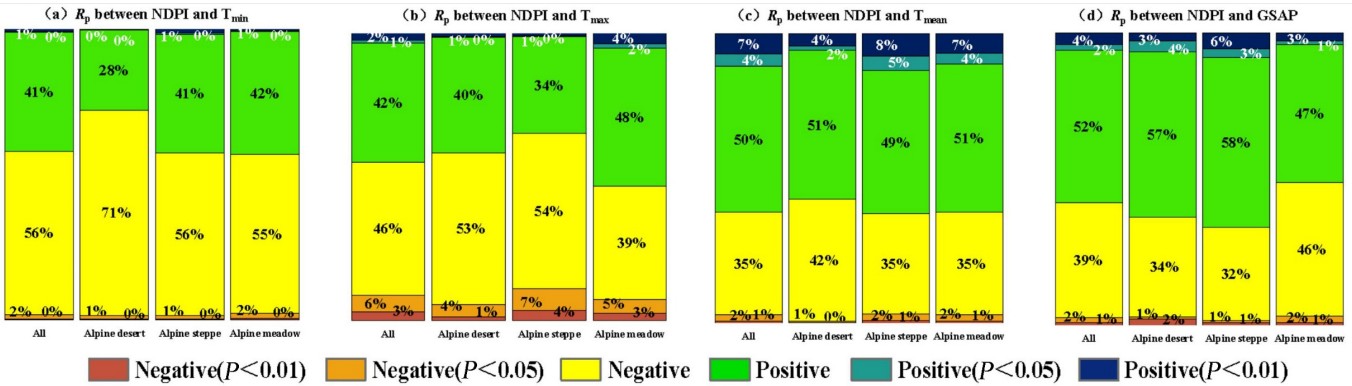

**Figure 5.** Area proportion of the partial correlation between NDPI and hydrothermal conditions for different grassland types.

An in-depth assessment was conducted to determine the sensitivity of NDPI to hydrothermal factors across all grid cells on the plateau (Figure 4e–h). As anticipated, even though spatial patterns of sensitivity and correlation varied, they predominantly exhibited identical signs in the majority of grid cells. The eastern region of the plateau predominantly displayed a negative sensitivity of NDPI to $T_{min}$, primarily falling below $12 \times 10^{-3}$ °C$^{-1}$ (Figure 4e). A significant portion, representing 63.66% of all grid cells, exhibited a positive sensitivity of NDPI to $T_{max}$ across the plateau, whereas the northeastern region demonstrated a more pronounced negative sensitivity, less than $-24 \times 10^{-3}$ °C$^{-1}$ (Figure 4f). In opposition, the northeastern region manifested a robust positive sensitivity of NDPI to $T_{mean}$, exceeding $12 \times 10^{-2}$ °C$^{-1}$. Conversely, for other regions of the plateau, the sensitivity of NDPI to $T_{mean}$ was notably subdued (Figure 4g). A substantial concentration of pixels, reflecting a high sensitivity greater than $6 \times 10^{-4}$ mm$^{-1}$ of NDPI to GSAP, was identified in the western region of the plateau. In contrast, other sections presented a relatively diminished sensitivity (Figure 4h).

### 3.3. Elevation-Dependent Differences in Hydrothermal Response and Sensitivity of NDPI

Our analysis highlights the pivotal role of elevation in modulating the response and sensitivity of grassland vegetation greenness to hydrothermal factors (Figure 6). This response and sensitivity of NDPI with respect to $T_{min}$, $T_{max}$, $T_{mean}$, and GSAP were elucidated at 10 m elevation intervals (Figure 6). Overall, the response and sensitivity of NDPI to hydrothermal factors distinctly fluctuated at elevations below 3000 m and above 5500 m. Notably, the partial correlation coefficient between NDPI and $T_{min}$ surged from $-0.26$ at an elevation of 970 m to a positive value ($R_p = 0.12$, $p > 0.05$) at 6260 m ($R^2 = 0.50$, $p < 0.05$) (Figure 6a). Conversely, the coefficient for NDPI and $T_{max}$ declined from a pronounced positive value ($R_p = 0.55$, $p < 0.05$) at 970 m to $-0.09$ at 6260 m (Figure 6a). The positive partial correlation coefficient between NDPI and $T_{mean}$ exhibited a unimodal trend across elevations, ranging from less than 2000 m to greater than 6000 m (Figure 6c). The correlation coefficient of NDPI with GSAP demonstrated a marginally declining trajectory ($p > 0.05$) (Figure 6d). As anticipated, although the sensitivity profile differed from correlation, they exhibited analogous trends, maintaining consistent signs with elevation in the majority of regions (Figure 6e–h).

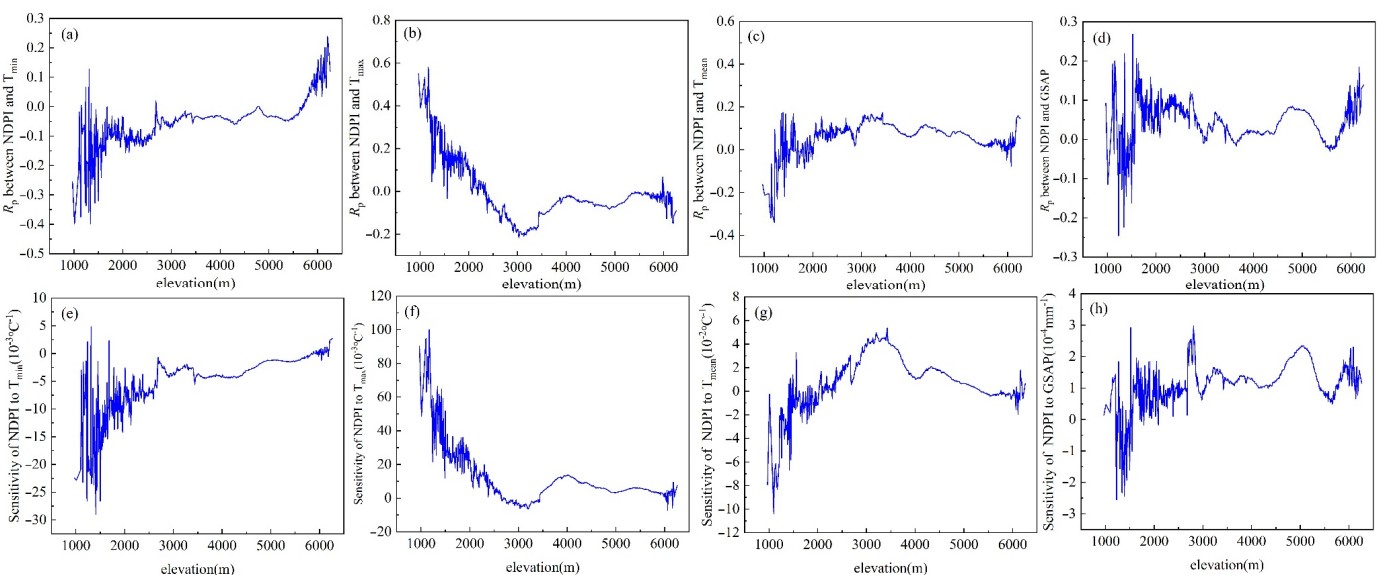

**Figure 6.** The response and sensitivity of NDPI to hydrothermal conditions in the peak season along the elevational gradient (at 10 m interval bins). In (**a**–**d**) the method of the partial correlation coefficient was used; in (**e**–**h**) the method of the regression coefficient was used.

## 4. Discussion

### 4.1. Estimation of NDPI and Its Hydrothermal Factors

The NDPI exhibited a rising trend during the peak season (Figure 2c), suggesting a consistent improvement in vegetation photosynthetic activity over the past 22 years in this region. This observation aligns with other studies utilizing NDVI data [7,16,17,20,41,42]. The increase in NDPI can be attributed to enhanced hydrothermal conditions observed locally over the past 22 years (Figure 3) and the introduction of several protective measures, such as the Protection and Construction of the National Ecological Security Shelter Zone (PCNESSZ) and the Returning Rangeland to Grassland (RRG) program. Additionally, ecological restoration initiatives like the Ecological Subsidy and Award System (ESAS) have been rolled out by governments since 2000 [43]. Nevertheless, a noteworthy decrease in NDPI was observed in 2.97% of the regions (Figure 2c,d), primarily spanning the Alpine steppe and Alpine desert areas. This aligns with findings from studies using NDVI data [7,17,20,40,41]. This declining trend likely stems from water scarcity resulting from the warming–drying climate [17]. While increased precipitation was recorded in these regions, elevated evapotranspiration due to rising temperatures might exacerbate the limitations on vegetation photosynthesis imposed by drought stress [44]. Notably, when juxtaposed with the grasslands of the Tibet Plateau, the Alpine grasslands demonstrate a more marked greening, potentially attributable to snowmelt [45].

### 4.2. Importance of Quantifying Hydrothermal Response and Sensitivity of NDPI

This study revealed that grassland NDPI generally exhibited positive responses to $T_{mean}$ and GSAP (Figure 4), implying that the greenness of Alpine grasslands could enhance if hydrothermal conditions trend towards being warmer and wetter. This finding aligns with previous reports [7,17,44]. In the eastern portion of the study area, NDPI primarily showed a positive response to $T_{mean}$ ($p < 0.05$) because temperature primarily restricts Alpine meadow vegetation growth compared to precipitation [17,46]. However, since 2000, GSAP has negatively influenced NDPI, indicating that excessive precipitation adversely affects vegetation growth due to exacerbated soil erosion and diminished soil organic matter [17,47]. This heightened precipitation also leads to reduced temperature and radiation, consequently inhibiting plant photosynthesis. From 2000 to 2021, the interannual variation of NDPI displayed negative correlations with both $T_{min}$ and $T_{max}$ across the majority of the plateau, suggesting that an increase in both maximum and minimum temperatures could decrease the greenness of Alpine grasslands. One potential reason

is that $T_{min}$ might impact seed germination and cause direct damage to vegetation cell structures [12,47,48]. The frozen soil water at $T_{min}$ temperatures could also restrict water uptake by plant roots [12,47]. Conversely, if $T_{min}$ rises, it could diminish freezing damage in plants, reduce seedling mortality, and enhance the photosynthetic capacity and growth rates, thereby extending the growing season [12]. $T_{max}$, to a degree, might indirectly cause a reduction in vegetation greenness by augmenting evaporation and respiration, consequently limiting water availability [49]. This could also heighten the risk of chlorophyll degradation and plant mortality in arid areas [50]. On the contrary, an increased $T_{max}$ might alleviate the cold temperature constraints on vegetation growth in relatively moist and cool ecosystems [49]. Interestingly, compared to this study. Notably, Shen et al. [12] discovered that summer vegetation greenness (July to August) on the Tibetan Plateau was strongly positively correlated with summer $T_{min}$ and negatively with $T_{max}$. This discrepancy arises because our study focuses on the growing season from May to September.

### 4.3. Vertical Functional Difference of Hydrothermal Factors on NDPI

A comparative analysis of hydrothermal factor effects on grassland vegetation greenness across various vertical gradient zones of the Plateau reveals distinct patterns in NDPI distribution's response and sensitivity to $T_{min}$, $T_{max}$, $T_{mean}$, and GSAP depending on elevation. There is an increasing positive response and heightened sensitivity of Alpine grasslands NDPI to $T_{min}$ as elevation rises on the Plateau ($p < 0.05$). One rationale behind this trend is that an elevation in $T_{min}$ may stimulate enhanced photosynthetic activity within plant thermal budgets at higher elevations and colder zones [51]. Conversely, there's a decreasing positive response and diminished sensitivity of NDPI to $T_{max}$ with increasing elevations ($p < 0.05$). This can primarily be attributed to the reduced water availability at higher elevations, potentially constraining or negating the favorable influence of $T_{max}$ on vegetation greening [16,42,44]. The findings demonstrate that NDPI's response and sensitivity to $T_{mean}$ across varying elevations largely depict unimodal patterns on the Tibetan Plateau, largely due to water availability constraints [42]. Both observations underscore that while temperature might offer optimal benefits, the effects of limited water availability could curtail vegetation growth. The data further reveal that NDPI's response and sensitivity to GSAP, when viewed in relation to elevation, suggest a subtle declining trend, aligning with the decreased water availability at greater elevations on the Tibetan Plateau. An alternative perspective posits that vegetation might manifest transient greenness influenced by environmental conditions, but extended stability, potentially driven by other factors such as radiation, $CO_2$, and nitrogen deposition [44,52], could counterbalance the impacts of changing hydrothermal conditions [42,47].

### 4.4. Uncertainties, Limitations and Future Perspectives

This study possesses inherent uncertainties and limitations. The primary sources of uncertainties stem from the quality and sources of data (notably, the remote sensing and meteorological datasets), as well as the employed analysis methods. Several notable limitations are as follows: Firstly, GSAP may not accurately represent water availability on the Tibetan Plateau; thus, the sensitivity of Alpine grassland vegetation greenness to GSAP variability could potentially be underestimated. Secondly, the study did not consider the potential lag effects of hydrothermal factors on greenness (for instance, the influence of hydrothermal factors on grassland greenness prior to the growth season). Thirdly, due to the absence of such conditions during the study period, the impacts of hydrothermal extremes on grassland greenness and the greenness's response and sensitivity to such extremes were not evaluated [53,54]. Lastly, given data constraints, the study did not delve into various dimensions such as the influence of radiation on greenness and hydrothermal conditions, human interventions like grazing, land use and socio-economic variables [7]. These will be incorporated in subsequent studies. Consequently, a more comprehensive study and profound analysis are required to address these uncertainties and limitations, thereby refining the findings. Additionally, attention should be directed

towards the sensitivity of grassland greenness to extreme hydrothermal events, especially heat-waves and droughts [20,37,54], in forthcoming research. Notwithstanding these limitations, the findings from this study stand to aid governments in crafting early warning systems against Alpine grassland degradation.

## 5. Conclusions

This study presents a thorough examination of the response and sensitivity of grassland vegetation greenness to contemporary hydrothermal conditions on the Tibetan Plateau, and it broadens sensitivity assessments of greenness along the vertical dimension. For the growing season spanning 2000–2021, both the response and sensitivity exhibited spatial heterogeneity. With elevation, the positive response and sensitivity of Alpine grassland vegetation greenness to minimum temperature increase markedly. In contrast, the response to maximum temperature exhibits an inverse relationship. The positive response and sensitivity of greenness to mean temperature, with respect to elevations, display unimodal patterns across the Tibetan Plateau. These insights are instrumental for evaluating the ecological repercussions on the Tibetan Plateau due to global climate shifts, especially in elevationally diverse regions. They also provide guidance in formulating grassland management strategies, ensuring the preservation of these delicate eco-climatic zones.

**Author Contributions:** Conceptualization, Y.W., X.X. and X.L.; methodology, Y.W. and B.S.; software, Y.W., L.H. and S.C.; validation, Y.W. and J.Z.; formal analysis, Y.W. and Y.L.; resources, Y.W., C.S., D.X. and H.L.; data curation, Y.W. and L.M.; writing—original draft preparation, Y.W. and B.S.; writing—review and editing, Y.W., C.S., M.L., H.L. and D.X.; supervision, X.X. and X.L.; project administration, X.X.; funding acquisition, X.X. All authors have read and agreed to the published version of the manuscript.

**Funding:** This study was supported by the National Key Research and Development Program of China (2021YFD1300500, 2021YFF0703904); the National Natural Science Foundation of China (32130070, 31971769, 41771205, 42101372); Special Funding for the Modern Agricultural Technology System from the Chinese Ministry of Agriculture (CARS-34); the Fundamental Research Funds Central Non-profit Scientific Institution (1610132021016). The Institute of General and Experimental Biology SB RAS (121030900138-8).

**Institutional Review Board Statement:** Not applicable.

**Informed Consent Statement:** Not applicable.

**Data Availability Statement:** The data presented in this study are available on request from the corresponding author. The data are not publicly available due to intellectual property.

**Acknowledgments:** We are grateful to many colleagues with the Hulunber Grassland Ecosystem Observation and Research Station, Institute of Agricultural Resources and Regional Planning, Chinese Academy of Agricultural Sciences (CAAS). Acknowledgment is given for the data support from "A Big Earth Data Platform for Three Poles (http://poles.tpdc.ac.cn/zh-hans/, accessed on 4 March 2023) and National Tibetan Plateau/Third Pole Environment Data Center (https://data.tpdc.ac.cn/home, accessed on 4 March 2023)".

**Conflicts of Interest:** Author Beibei Shen was employed by the company Aerospace Science and Industry (Beijing) Spatial Information Application Co., Ltd. The remaining authors declare that the research was conducted in the absence of any commercial or financial relationships that could be construed as a potential conflict of interest.

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
