# Peer review of "Elevation-Dependent Contribution of the Response and Sensitivity of Vegetation Greenness to Hydrothermal Conditions on the Grasslands of Tibet Plateau from 2000 to 2021"

_remotesensing, doi:10.3390/rs16010201_

Round 1
Reviewer 1 Report
Comments and Suggestions for Authors
The manuscript “Elevation-dependent contribution of the response and sensitivity of vegetation greenness to hydrothermal conditions on the grasslands of Tibet Plateau from 2000 to 2021” (Yatang Wu et al.) is devoted to a problem that is both theoretical and practical, and it is interesting. The authors, based on the Normalized Difference Phenology Index (NDPI) and four hydrothermal conditions (minimum/maximum/mean temperature and precipitation) during the growing season, analyzed the response and sensitivity of the NDPI to hydrothermal conditions with at Tibet Plateau elevations between 2000 and 2021. Landscape reference taken from the Chinese Grassland Classification System: Alpine meadows, Alpine steppes, and Alpine desert are the three main grassland types distributed in this region. The compiled maps reflect both the greenness of vegetation and the individual hydroclimatic characteristics well.
My comments:
1) The text only provides a statement of the facts obtained, including where and what indicators have certain characteristics. For example, the most widespread low values (mean NDPI <0.3) with poorer vegetation photosynthetic activity are mainly distributed in the northwestern part of the plateau, and in contrast, NDPI showed higher values (mean NDPI > 0.6) with greater vegetation photosynthetic activity and mainly concentrated in the eastern and northeastern parts of the plateau, or Tmin showed lower values in the northwest part, while the highest Tmax occurred in the northern part of the plateau. The same is true for the characterization of correlations between NDPI and hydroclimatic indicators. Regional features of climate processes are not analyzed. Without this, understanding the dynamics of NDPI is impossible. It is necessary for authors to submit their versions.
2) In my view, Figure 5 fails to demonstrate the impact of elevation on the response and sensitivity of vegetation greenness. I believe it should stay in the text.
3) In the discussion, it would be good to consider changing trends in the development of characteristics of mountain meadows in other mountain regions such as the Alps, Andes, and Caucasus. Your work's significance could be appreciated by readers by doing this.
In general, I believe that the manuscript needs to be improved.
Author Response
Response to reviewer 1
Comments and Suggestions for Authors
The manuscript“Elevation-dependent contribution of the response and sensitivity of vegetation greenness to hydrothermal conditions on the grasslands of Tibet Plateau from 2000 to 2021” (Yatang Wu et al.) is devoted to a problem that is both theoretical and practical, and it is interesting. The authors, based on the Normalized Difference Phenology Index (NDPI) and four hydrothermal conditions (minimum/maximum/mean temperature and precipitation) during the growing season, analyzed the response and sensitivity of the NDPI to hydrothermal conditions with at Tibet Plateau elevations between 2000 and 2021. Landscape reference taken from the Chinese Grassland Classification System: Alpine meadows, Alpine steppes, and Alpine desert are the three main grassland types distributed in this region. The compiled maps reflect both the greenness of vegetation and the individual hydroclimatic characteristics well.
Response:First of all, thank you for your careful and professional review. Your comments are of great significance to the improvement of our original manuscript. We studied and read intensively your comments one by one and made detailed answers below. The revised contents have been marked in red in the revised manuscript.
Major comments:
#1 The text only provides a statement of the facts obtained, including where and what indicators have certain characteristics. For example, the most widespread low values (mean NDPI <0.3) with poorer vegetation photosynthetic activity are mainly distributed in the northwestern part of the plateau, and in contrast, NDPI showed higher values (mean NDPI > 0.6) with greater vegetation photosynthetic activity and mainly concentrated in the eastern and northeastern parts of the plateau, or Tmin showed lower values in the northwest part, while the highest Tmax occurred in the northern part of the plateau. The same is true for the characterization of correlations between NDPI and hydroclimatic indicators. Regional features of climate processes are not analyzed. Without this, understanding the dynamics of NDPI is impossible. It is necessary for authors to submit their versions.
Response 1:Thank you for this very important comment. The Theil-Sen (TS) method and the non-parametric rank-based Mann–Kendall (MK) test was used to identify the trends and significance of NDPI in this study. Fig. 2c reveals the spatial distribution of the overall trends in annual grassland NDPI within the study area during the peak season over the past 22 years. Fig. 3 presents the spatial distribution and trends of Tmin, Tmax, Tmean, and GSAP.
#2 In my view, Figure 5 fails to demonstrate the impact of elevation on the response and sensitivity of vegetation greenness. I believe it should stay in the text.
Response 2: Thank you for your guidance on this important question. Figure 5 shows the areas percentage of partial correlation between NDPI and Tmin, Tmax, Tmean and GSAP across the three different grassland types, respectively. Figure 6 shows the response and sensitivity of NDPI to hydrothermal conditions in the peak season along the elevational gradient (at 10 m interval bins) to investigate the effects of hydrothermal factors on grassland vegetation greenness distributed in the different vertical gradient zones of Plateau.
#3 In the discussion, it would be good to consider changing trends in the development of characteristics of mountain meadows in other mountain regions such as the Alps, Andes, and Caucasus. Your work's significance could be appreciated by readers by doing this. In general, I believe that the manuscript needs to be improved.
Response 3: Thanks for your valuable suggestions. According to your suggestion, we have reviewed the relevant references and added the comparisons between grasslands of Tibet Plateau and other grasslands in other mountain regions such as the Alps in the revised manuscript as below. “Notably, when juxtaposed with the grasslands of the Tibet Plateau, the Alpine grasslands demonstrate a more marked greening, potentially attributable to snowmelt [46].”
References
- Choler, P.; Bayle, A.; Carlson, B.Z.; Randin, C.; Filippa, G.; Cremonese, E.The tempo of greening in the European Alps: Spatial variations on a common theme.Global Change Biology. 2021, 27, 5614-5628.https://doi.org/10.1111/gcb.15820.

Reviewer 2 Report
Comments and Suggestions for Authors
1. There have been a large number of studies on the response of vegetation to climate change at regional and global scales. The authors need to systematically review the relevant studies in the introduction section and clarify the innovation of this study.
2. In the discussion section, the authors need to systematically compare the results of this study with those of previous studies.
3. NDPI is not very common index for vegetation greenness, LAI product also needs to be included
4. In addition to temperature and precipitation, radiation is also an important factor, need to be considered
5. The quality of the figures were quite bad, supposed to be reproduced
6. The spatial distribution grassland area is varied between 2000 and 2021, it is incorrect to use identical grassland map.
Comments on the Quality of English LanguageExtensive editing of English language required
Author Response
Response to reviewer 2
Comments and Suggestions for Authors
Thank you very much for your constructive comments on the original manuscript. We admire your profound professional knowledge and tireless attitude. We have tasted your comments word for word and made detailed modifications to the original manuscript. The revised contents include the major revision of charts and logic, etc. Additionally, we have carefully revised the mistakes in the grammar and extensively edited English language of the manuscript. We studied and read intensively your comments one by one and made detailed answers below. The revised contents have been marked in red in the revised manuscript.We believe that your comments will greatly improve the quality of our manuscript and help it be published as soon as possible.
#1 There have been a large number of studies on the response of vegetation to climate change at regional and global scales. The authors need to systematically review the relevant studies in the introduction section and clarify the innovation of this study.
Response 1: Thanks for your kind suggestion. Your opinion is of great significance to us. According to your suggestion, we have reviewed the relevant references and enriched the content in the Introduction section and systematically reviewed the relevant studies. For example,we have optimized the fourth paragraph in the Introduction and clarified the innovation of this study as below.“Numerous studies have investigated the response of vegetation greenness to climate change at both regional and global scales. For example, Xu et al. [28] demonstrated that NDVI changes were significantly correlated with annual temperature (R= 0.52, P < 0.01) across China but not with annual precipitation (P > 0.1). Additionally, correlations between vegetation greenness changes and both temperature and precipitation were found to be significant at a regional scale (P < 0.001). However, many of these studies primarily focus on the annual mean temperature, neglecting the potential influences of minimum and maximum temperatures during the growing season. Previous research has also highlighted the significant impact of elevation on the variations in Alpine vegetation greenness in response to hydrothermal conditions on the Tibetan Plateau [11, 14–16]. Specifically, An et al. [14] identified a pronounced elevation-dependent relationship between vegetation greenness and temperature during the growing season (May–Sep) from 2000–2016. Furthermore, Wang et al. [16] pointed out a mismatch between the elevational variation rate of NDVI and hydrothermal conditions. At altitudes above 2400 m, temperature predominantly influenced the elevational shifts of NDVI isolines, whereas precipitation was the dominant factor below 2400 m. Wang et al. [16] also observed that the drought response (SPI/SPEI) of the Enhanced Vegetation Index (EVI) significantly diminished with increasing elevation (P < 0.001). Despite these findings, the specific adjustion of elevation on the response and sensitivity of grassland vegetation greenness to minimum/maximum temperatures during the Tibetan Plateau's growing season remains unexplored. This study seeks to address this gap”
#2 In the discussion section, the authors need to systematically compare the results of this study with those of previous studies.
Response 2: Thanks for your valuable suggestions. Your opinion is of great significance to us. According to your suggestion, We have carefully reviewed relevant reference and rewritten and revised the discussion in detail and systematically compared the results of this study with those of previous studies as below. “The NDPI exhibited a rising trend during the peak season (Figure 2c), suggesting a consistent improvement in vegetation photosynthetic activity over the past 22 years in this region. This observation aligns with other studies utilizing NDVI data [7, 16–17, 20, 42–43]. ” and “Nevertheless, a noteworthy decrease in NDPI was observed in 2.97% of the regions (Figure 2c and 2d), primarily spanning the Alpine steppe and Alpine desert areas. This aligns with findings from studies using NDVI data [7, 17, 20, 41–42]. ” and “Interestingly, compared to this study, notably, Shen et al. [12] discovered that summer vegetation greenness (July to August) on the Tibetan Plateau was strongly positively correlated with summer Tmin and negatively with Tmax. This discrepancy arises because our study focuses on the growing season from May to September.”
#3 NDPI is not very common index for vegetation greenness, LAI product also needs to be included.
Response 3: We strongly agree with your suggestions. We believe that the above suggestions will be of great significance for our future research. NDVI is the most commonly used indicator to measure vegetation greenness. However, The accuracy of NDVI is subject to scrutiny due to its tendency to saturate in areas of dense vegetation and its susceptibility to soil backgrounds, canopy brightness, and shadows when coverage falls below 50%. Additionally, atmospheric disturbances, such as aerosols, frequently introduce noise into images generated from red and near-infrared bands. It's significant to understand that grasslands typically exhibit lower vegetation coverage and smaller canopy dimensions, coupled with increased spatial heterogeneity. This often leads to satellite imagery comprising more bare soil pixels relative to other ecosystems. Nevertheless, Wang et al. [18] introduced the NDPI, an innovative vegetation index designed to discern the difference between green vegetation and soil backgrounds while attenuating this differentiation. Employing a weighted shortwave infrared (SWIR) band in lieu of the red band in NDVI, NDPI is sensitive to vegetation water content, enabling it to track variations in canopy water content [18]. Crucially, NDPI merges these two functionalities into a singular VI without compromising its sensitivity to vegetation greenness [18]. Xu et al. [19] demonstrated NDPI's superior capability in estimating aboveground fresh biomass across expansive grassland regions with pronounced spatial heterogeneity, particularly in diminishing the interference of soil background in Inner Mongolia grasslands. In conclusion, NDPI presents itself as a promising tool for gauging the greenness of Alpine grassland vegetation.
References
- Wang, C.; Chen, J.; Wu, J.; Tang, Y.H.; Shi, P.J.; Black, T.A.; Zhu, K. A snow-free vegetation index for improved monitoring of vegetation spring green-up date in deciduous ecosystems. Remote Sens. Environ. 2017, 196, 1-12. https://doi.org/10.1016/j.rse.2017.04.031.
- Xu, D.W.; Wang, C.; Chen, J.; Shen, M.G.; Shen, B.B.; Yan, R.R.; Li, Z.W.; Karnieli, A.; Chen, J.Q.; Yan, Y.C.; et al. The superiority of the normalized difference phenology index (NDPI) for estimating grassland aboveground fresh biomass. Remote Sens. Environ. 2021, 264, 112578. https://doi.org/10.1016/j.rse.2021.112578.
#4 In addition to temperature and precipitation, radiation is also an important factor, need to be considered.
Response 4: Thank you for your guidance on this important question. Your opinion is of great significance to us. We have carefully reviewed relevant references and strongly agree with your suggestions. In the revised manuscript, We are currently unable to examine the influence of radiation on greenness and hydrothermal conditions due to the limitations in the availability of radiation data from 2000 to 2021 . We have added uncertainties and limitations of the study in the Discussion as below. “Lastly, given data constraints, the study did not delve into various dimensions such as the influence of radiation on greenness and hydrothermal conditions, human interventions like grazing, land use and socio-economic variables [7]. These will be incorporated in subsequent studies.” .
#5 The quality of the figures were quite bad, supposed to be reproduced.
Response 5: Thanks for your valuable suggestions. Your opinion is of great significance to us. The reason for the quite bad quality of the figures may be that the pixels were compressed when we pasted the image into the manuscript. In the revised manuscript, we make every effort to improve and reproduce all the figures.
#6 The spatial distribution grassland area is varied between 2000 and 2021, it is incorrect to use identical grassland map.
Response 6: We strongly agree with your suggestions. Indeed, the grassland area changes every year, but this change is very small and it can be ignored. Additionally, The grassland type map currently lacks remote sensing products, and the update lag cannot achieve annual changes. Therefore,we have no choice but to use the identical grassland map. But thank you very much for the comments and suggestions. We have added the suggestions of the study in the Discussion as below. “Lastly, given data constraints, the study did not delve into various dimensions such as the influence of radiation on greenness and hydrothermal conditions, human interventions like grazing, land use and socio-economic variables [7]. These will be incorporated in subsequent studies.”
#7 Comments on the Quality of English Language Extensive editing of English language required.
Response 7: We sincerely admire your professionalism and thank you for your careful and professional comments on this manuscript. We have carefully revised the mistakes in the grammar and extensively edited English language of the manuscript. We are very grateful to you for your many constructive comments on our manuscript, which will have an inestimable effect on improving the manuscript. We also look forward to the early publication of the manuscript.

Reviewer 3 Report
Comments and Suggestions for Authors
See attachment

Comments on the Quality of English LanguageSome sentences seem incomplete and some words are missing
Author Response
Response to reviewer 3
Comments and Suggestions for Authors
Response: We sincerely admire your professionalism and thank you for your careful and professional comments on this manuscript. We study your views and revise the manuscript one by one. We believe that under your guidance, the improvement of manuscript will be constructive.
Major comments:
#1 You wrote: Here, we remedy this deficiency by: 1) quantifying Tibetan Plateau grasslands vegetation greenness response or sensitivity to interannual hydrothermal conditions (mini-mum/maximum/mean temperature and precipitation) variations from 2000 to 2021;….
My comments: After going through your results, I could only see the percent of greenness and some statistics such as P-values. Would you please explain what you mean by quantification?
Response 1: Thank you very much for the comments and suggestions. Your opinion is of great significance to us. After careful consideration, We also believe that the term “quantification” is not appropriate.We have rewritten this paragraph, and this misstatement has been deleted.We have replaced “quantifying” on the original manuscrip with “examining”, which is more accurate. We have checked the whole manuscript by and rectified all similar problems.
#2 You wrote:2.3. Datasets processing The spatial distribution map of the mean grassland NDPI during the peak of the growing season for the period from 2000 to 2021 were obtained by way of calculating average value for the whole study area in each of the past 22 years.
My comments: What is the peak of the growing season for this area? Were the MODIS data sets over the 22 years, the same dates and time of the day? If there were taken at different dates and time, results may be affected.
Response 2: Thank you for your guidance on this important question. The peak of the annual growing season are inconsistent for this area. We collected the 8-day NDPI data from May to September of each year. Ultimately, the maximum-value compositing method was utilized to compile the 8-day NDPI data, resulting in a synthesized value for the entire growing season. Therefore, the peak of the growing season of each year is not fixed.
#3 You wrote:
The Digital Elevation Model (DEM) database, consisting of the global 90 m resolution (downloaded from http://srtm.csi.cgiar.org), were accessed to characterize elevation over the Tibetan Plateau, which derived from Shuttle Radar Topography Mission (SRTM) images. To match the valid data, the DEM was resampled to 500 m resolution, and the projection type was defined as Albers Equal Area projection for this study. Ultimately, these grids were utilized to study.
My comments: What software did you use to resample 90 m resolution into 500 m ground resolution?My comments: How did you come up with 10-m elevation intervals. See: 3.3. Elevation-dependent differences in hydrothermal response and sensitivity of NDPI We also found that key role of elevation in regulating the response and sensitivity of grassland vegetation greenness to hydrothermal factors (Figure 6). The response and sensitivity of NDPI to Tmin, Tmax, Tmean and GSAP, respectively, were also showed at 10-m elevation intervals.
Response 3: Thank you for your guidance on this important question. The Digital Elevation Model (DEM) database, consisted of the global 90m resolution datasets, and were accessed to characterize the altitudes of the Tibetan Plateau. To match the valid data, the DEM was resampled to 500m resolution in this study through the use of ArcGIS (ArcGIS 10.7) software (ESRI, Inc., Redlands, CA, USA). According to Figure 1, Average of pixels along the elevational gradient (at 10 m interval bins) by the homemade scripts written within the statistical software package in python (3.11.3).
General observations
#4 It was not clear to me what type of statistical test you used to make your inferences about the difference in greenness sensitivity of different aspects (the orientation of slope) in your study area? The significance of differences cannot rely on correlation results and P-values. Remember that P-value for a large sample size used in Remote Sensing is generally small and can be misleading.
Response: Thank you for this very important comment. The inter-annual change of the NDPI was analyzed using the slope model calculated for each pixel across the study area by the homemade scripts written within the statistical software package in R (4.1.2). The NDPI response to temperature was determined by a pixel-by-pixel partial correlations between the spatial distribution of NDPI and temperature by the homemade scripts written within the statistical software package in R (4.1.2). Therefore, there are a total of 22 samples per pixel from 2000 to 2021. Both slope and correlation are performed by a pixel-by-pixel. We added this content as below.“Trend and significance of NDPI by a pixel-by-pixel were discerned using the Theil-Sen (TS) method combined with the non-parametric rank-based Mann–Kendall (MK) test [40–41]” and “To evaluate the interannual variations in the maximum greenness of grassland vegetation in response to Tmin during the growing season, partial correlation coefficients were computed between NDPI by a pixel-by-pixel and Tmin, considering Tmax, Tmean, and GSAP as control variables [12]. The apparent sensitivity of NDPI to Tmin was quantified by the coefficient derived from multiple linear regressions, where NDPI by a pixel-by-pixel was regressed against Tmin, Tmax, Tmean, and GSAP [12]. ”
#5 I could have learnt more about your findings if the points you raised in “4.4. Uncertainties, limitations and future perspectives”were actually included in this study. I am wondering if higher resolution images, 30 m instead of 500 m, would have yielded different results.
Response 5: Thank you very much for the suggestion. Due to the limitations in availability of high resolution meteorological data, analysing the response and sensitivity of NDPI with 30 m spatial resolution to meteorological conditions with coarse resolution on the plateau scale is inappropriate. But, Using NDPI with a 30 meter spatial resolution to analyze its spatial pattern and changes will be more accurate and it will also produce different results compared to the 500 m spatial resolution. We believe that the above suggestions will be of great significance for our future research.
#6 Comments on the Quality of English Language: Some sentences seem incomplete and some words are missing.
Response 6: We sincerely admire your professionalism and thank you for your careful and professional comments on this manuscript. We have carefully revised the mistakes in the grammar and extensively edited English language of the manuscript. We are very grateful to you for your many constructive comments on our manuscript, which will have an inestimable effect on improving the manuscript. We also look forward to the early publication of the manuscript.

Round 2
Reviewer 1 Report
Comments and Suggestions for Authors
I took care to review the authors' responses to comments and the revised version of the manuscript. The responses to the comments have left me satisfied. The authors' work in correcting and improving the manuscript is something I admire. I wish the authors continued success with their work.
Author Response
Response to reviewer 1
Comments and Suggestions for Authors
I took care to review the authors' responses to comments and the revised version of the manuscript. The responses to the comments have left me satisfied. The authors' work in correcting and improving the manuscript is something I admire. I wish the authors continued success with their work.
Response:Thank you very much for your constructive comments on the original manuscript. We admire your profound professional knowledge and tireless attitude. Wishing you good health and smooth work.

Reviewer 2 Report
Comments and Suggestions for Authors
The authors did not answer questions and revise manuscript well.
1. In the past 20 years, the spatial distribution of grassland varied, and it is inappropriate to ignore this variation.
2. Radiation is a very important meteorological factor, without considering this factor the result is not reliable.
3. NDPI is not a common vegetation index, leaf area index can indicate vegetation greenness more appropriate.
4. The mean value is a commonly used index in previous studies, so it is necessary to supplement the analysis of the mean instead of only using the maximum and minimum value.
Comments on the Quality of English LanguageExtensive editing of English language required
Author Response
Response to reviewer 2
Comments and Suggestions for Authors
The authors did not answer questions and revise manuscript well.
Response : First of all, thank you for your careful and professional review. Your comments are of great significance to the improvement of our original manuscript. We studied and read intensively your comments one by one and made detailed answers below. The revised contents have been marked in red in the revised manuscript.
Major comments:
#1 In the past 20 years, the spatial distribution of grassland varied, and it is inappropriate to ignore this variation.
Response 1: Thank you very much for the comments. Grassland types were derived from the 1 : 1 000 000 Chinese digital grassland classification map provided by the China Resource and Environmental Science Data Center (https://www.resdc.cn/, last access: 8 February 2023). This dataset, generated through field surveys in the 1980s and supplemented by satellite and aerial imagery, is the most detailed grassland-type map available. To facilitate comparison with other study estimates, we regrouped the grassland types into three categories, i.e., Alpine meadow, Alpine steppe, and Alpine desert, and resampled this regrouped vector to a grid with 500 m spatial resolution. The following paper and I use the same grassland type (Qin., et al., 2018; Zeng., et al., 2019; Li., et al., 2020; Gao., et al., 2020; Duan., et al., 2021; Yu., et al., 2021; Cheng et al., 2023; Zhang et al., 2023). At present, however, there is no analysis and study using grassland type maps that change every year.
References
Qin X, Sun J, Wang X. Plant coverage is more sensitive than species diversity in indicating the dynamics of the above-ground biomass along a precipitation gradient on the Tibetan Plateau[J]. Ecological Indicators, 2018, 84: 507-514.
Zeng N, Ren X, He H, et al. Estimating grassland aboveground biomass on the Tibetan Plateau using a random forest algorithm[J]. Ecological Indicators, 2019, 102: 479-487.
Li P, Hu Z, Liu Y. Shift in the trend of browning in Southwestern Tibetan Plateau in the past two decades[J]. Agricultural and Forest Meteorology, 2020, 287: 107950.
Gao X, Dong S, Li S, et al. Using the random forest model and validated MODIS with the field spectrometer measurement promote the accuracy of estimating aboveground biomass and coverage of alpine grasslands on the Qinghai-Tibetan Plateau[J]. Ecological Indicators, 2020, 112: 106114.
Duan H, Xue X, Wang T, et al. Spatial and temporal differences in alpine meadow, alpine steppe and all vegetation of the Qinghai-Tibetan Plateau and their responses to climate change[J]. Remote Sensing, 2021, 13(4): 669.
Yu H, Wu Y, Niu L, et al. A method to avoid spatial overfitting in estimation of grassland above-ground biomass on the Tibetan Plateau[J]. Ecological Indicators, 2021, 125: 107450.
Cheng M, Wang Y, Zhu J, et al. Precipitation dominates the relative contributions of climate factors to grasslands spring phenology on the Tibetan Plateau[J]. Remote Sensing, 2022, 14(3): 517.
Zhang H, Tang Z, Wang B, et al. A 250 m annual alpine grassland AGB dataset over the Qinghai–Tibet Plateau (2000–2019) in China based on in situ measurements, UAV photos, and MODIS data[J]. Earth System Science Data, 2023, 15(2): 821-846.
#2 Radiation is a very important meteorological factor, without considering this factor the result is not reliable.
Response 2: Thank you for your guidance on this important question. Your opinion is of great significance to us. However, it is difficult to obtain solar radiation data or sunshine hours data at such a long time scale (2000-2021) on the Tibet Plateau. So far, I am unable to obtain these data with all my best efforts due to meteorological data including sunshine hours is no longer free and publicly available on the Tibet Plateau.
#3 NDPI is not a common vegetation index, leaf area index can indicate vegetation greenness more appropriate.
Response 3: Thank you very much for the comments. NDVI is the most commonly used indicator to measure vegetation greenness. However, NDPI is superior to NDVI in estimating aboveground fresh biomass across expansive grassland regions with pronounced spatial heterogeneity, particularly in diminishing the interference of soil (Xu et al. 2021). NDPI is superior to NDVI and EVI in Establishment of a Random Forest Model for grassland aboveground biomass (Xu et al. 2021). There are currently no published articles proving that NDPI is superior to LAI. However, our team's extensive in-situ samples on the Tibet Plateau can indirectly prove that NDPI is superior to LAI in constructing gross primary productivity (GPP) models (Figure 1). We conducted a large amount of in-situ data collection on the Tibet Plateau including photosynthetically active radiation (PAR) (Figure 2 and Figure 3).
Figure 1. Survey of key photosynthetic parameters in grasslands on the Tibet Plateau from 2022 to 2023.
The experiment selected clear and cloudless weather (July to August, 2022-2023) to measure the PAR components of the sample canopy, and used the AccuPAR LP-80 (produced by LI-COR company in the United States) plant canopy analyzer to measure the instantaneous photosynthetic effective radiation (PAR) components of the sample canopy. This instrument consists of two parts: a data collector and a probe rod. The 80 photosensitive sensors on the probe rod can measure the photosynthetically active radiation in the 400-700nm wavelength range, in units of μ Mol/m2/s. After being weakened by the atmosphere and reaching above the canopy of the sample plot, a portion of the PAR is reflected back into the atmosphere by the vegetation canopy during transmission, while the remaining PAR reaches the soil surface at the bottom of the vegetation canopy in the form of transmission after being reflected and absorbed by the canopy leaves layer by layer; Part of the PAR that reaches the soil surface is absorbed by the soil, while another part is reflected by the soil and reabsorbed by the vegetation. Only the PAR absorbed by the vegetation canopy contributes to the accumulation of dry matter in crops. Therefore, each time the four components of PAR are measured, namely canopy incident PAR ↓ AC, reflected PAR ↑ AC, transmitted PAR ↓ BC, and soil reflection PAR ↑ BC (Figure 2 and Figure 3).
Figure 2. PAR measurement principle diagram.
Figure 3. Author collects PAR data.
The ratio of photosynthetic effective radiation absorbed by vegetation (FAPAR) is the ratio of photosynthetic effective radiation absorbed by vegetation to total photosynthetic effective radiation. FAPAR=(PAR↓AC-PAR↑AC)-(PAR↓BC -PAR↑BC)/PAR↓AC。We collected a total of 585 FAPAR in-situ samples (Figure 4).
Figure 4. PAPAR distribution map.
I constructed gross primary productivity (GPP) models by establishing the relationship between FAPAR and four vegetation indexs including NDVI, LAI, EVI, NDPI (Table 1). The results indicate that the relationship between NDPI and PAPAR is superior to NDVI, LAI, and EVI in both linear and nonlinear aspects. Vegetation greenness characterizes vegetation photosynthetic activity and is highly correlated with FAPAR. Through in-depth analysis of in-situ data, we have demonstrated that NDPI is more superior than LAI in terms of selecting photosynthetic parameters in the Alpine grassland. Based on the above results, in conclusion, NDPI presents itself as a promising tool for gauging the greenness of Alpine grassland vegetation. In addition, the Innovation and highlight of this study is the use of NDPI to study the greenness of Alpine grasslands.
Table 1. The relationship between FAPAR and four vegetation indexs.
References
Xu, D.W.; Wang, C.; Chen, J, et al. The superiority of the normalized difference phenology index (NDPI) for estimating grassland aboveground fresh biomass. Remote Sens. Environ. 2021, 264, 112578.
Ding L, Li Z, Shen B, et al. Spatial patterns and driving factors of aboveground and belowground biomass over the eastern Eurasian steppe[J]. Science of The Total Environment, 2022, 803: 149700.
#4 The mean value is a commonly used index in previous studies, so it is necessary to supplement the analysis of the mean instead of only using the maximum and minimum value.
Response 4: Thank you very much for the comments.The highlight and innovate of this study is to explore the response and sensitivity of NDPI to minimum and maximum temperatures. Therefore, it is necessary to compare with previous research on minimum and maximum temperatures. We have demonstrated in the manuscript that the response of greenness to mean temperature and precipitation is consistent with previous studies. Please review as below “ This study revealed that grassland NDPI generally exhibited positive responses to Tmean and GSAP (Figure 4), implying that the greenness of Alpine grasslands could enhance if hydrothermal conditions trend towards being warmer and wetter. This finding aligns with previous reports [7, 17, 45].”
#5 Extensive editing of English language required.
Response 5: We sincerely admire your professionalism and thank you for your careful and professional comments on this manuscript. We have carefully revised the mistakes in the grammar and extensively edited English language of the manuscript. In addition, this paper has been sent to relevant institutions for polishing. We are very grateful to you for your many constructive comments on our manuscript, which will have an inestimable effect on improving the manuscript. We also look forward to the early publication of the manuscript.

Reviewer 3 Report
Comments and Suggestions for Authors
You wrote:
Remote sensing (RS) offers the benefits of efficiency, extensive coverage, and diverse information, establishing itself as the most effective tool for monitoring vegetation greenness dynamics at large scales.
My comments: Did you mean large extent or large spatial scale? The word large scale may mean a small area that has more detail. Thus, higher resolution.
You still did not clarify "peak season". Does it mean July?
Comments on the Quality of English LanguageNone
Author Response
Response to reviewer 3
Comments and Suggestions for Authors
Response: First of all, thank you for your careful and professional review. Your comments are of great significance to the improvement of our original manuscript. We studied and read intensively your comments one by one and made detailed answers below. The revised contents have been marked in red in the revised manuscript.
Major comments:
#1 Remote sensing (RS) offers the benefits of efficiency, extensive coverage, and diverse information, establishing itself as the most effective tool for monitoring vegetation greenness dynamics at large scales.
My comments: Did you mean large extent or large spatial scale? The word large scale may mean a small area that has more detail. Thus, higher resolution.
Response 1: Thank you very much for the comments. I mean is large range. In the revised manuscript, we have replaced “range” with “scale”.
#2 You still did not clarify "peak season". Does it mean July?
Response 2: Thank you very much for the comments. The "peak season" refers to July to August. According to the graph below, it can be seen that the growth season changes in a parabolic shape. NDPImax is the vertex of a parabola. The vertex also represents the peak of the growing season at each location. The vertex of the annual growing season are inconsistent for this area.
